# Forecasting Renewable Energy Generation with Machine Learning and Deep Learning: Current Advances and Future Prospects

Natei Ermias Benti *, Mesfin Diro Chaka and Addisu Gezahegn Semie *

Computational Data Science Program, College of Natural and Computational Sciences, Addis Ababa University, Addis Ababa P.O. Box 1176, Ethiopia
* Correspondence: natei.ermias@aau.edu.et (N.E.B.); addisu.semie@aau.edu.et (A.G.S.)

**Abstract:** This article presents a review of current advances and prospects in the field of forecasting renewable energy generation using machine learning (ML) and deep learning (DL) techniques. With the increasing penetration of renewable energy sources (RES) into the electricity grid, accurate forecasting of their generation becomes crucial for efficient grid operation and energy management. Traditional forecasting methods have limitations, and thus ML and DL algorithms have gained popularity due to their ability to learn complex relationships from data and provide accurate predictions. This paper reviews the different approaches and models that have been used for renewable energy forecasting and discusses their strengths and limitations. It also highlights the challenges and future research directions in the field, such as dealing with uncertainty and variability in renewable energy generation, data availability, and model interpretability. Finally, this paper emphasizes the importance of developing robust and accurate renewable energy forecasting models to enable the integration of RES into the electricity grid and facilitate the transition towards a sustainable energy future.

**Keywords:** accurate predictions; deep learning; energy management; machine learning; renewable energy forecasting

## 1. Introduction

Renewable energy research and development have gained significant attention due to a growing demand for clean and sustainable energy in recent years [1,2]. In the fight to cut greenhouse gas emissions and slow down climate change, renewable energy is essential [3–5]. In addition, renewable energy sources (RES) offer several advantages, including a reduction in energy dependence on foreign countries, job creation, and the potential for cost savings [1,6]. However, the inherent variability and uncertainty of RES present a significant challenge for the widespread adoption of renewable energy [7,8]. For example, wind energy generation is heavily influenced by the weather, which can be unpredictable and difficult to forecast accurately [9,10]. Similarly, solar energy generation is influenced by factors such as cloud cover and seasonal changes in sunlight [11]. The high variability and uncertainty of renewable energy generation make it challenging to integrate RES into the power grid efficiently [12].

One approach to addressing this challenge is to develop accurate forecasting models for renewable energy generation. Accurate forecasting models can help minimize the negative impact of the variability and uncertainty of renewable energy generation on the power grid. For decades, energy generation has been predicted using traditional forecasting models, such as statistical and physical models [13]. However, statistical models such as the autoregressive integrated moving average (ARIMA) method have limitations in their ability to handle complex nonlinear relationships and the high-dimensional nature of renewable energy data [14]. Physical models, such as numerical weather prediction (NWP) models and solar radiation models, are widely used for renewable energy forecasting. NWP models

use atmospheric data to predict wind speed and direction, while solar radiation models use cloud cover and atmospheric conditions to predict solar irradiance. However, these models have limitations due to the complexity of the Earth's atmosphere and the inherent uncertainty in weather forecasting. Improving these models' accuracy requires ongoing research and development to address these limitations [15,16]. Promising approaches to address the limitations of traditional forecasting models involve the utilization of ML and DL algorithms [14]. ML and DL algorithms can learn complex nonlinear relationships from immense quantities of information, making them suitable for handling the high-dimensional nature of renewable energy data. Moreover, they can handle a wide range of input data types, including time series data, meteorological data, and geographical data.

Many researchers have looked at the application of ML and DL algorithms for the forecasting of solar radiation, a significant element influencing the output power of solar systems [17]. For instance, Voyant et al. suggested the use of hybrid models (HMs) to enhance prediction performance after discovering that SVR, SVM, ARIMA, and ANN are the superior approaches for forecasting solar radiation [18]. Huertas et al. demonstrated that the HM with SVM outperformed single predictor models in terms of improving forecasts of solar radiation [19]. In comparison to other models, Gürel et al. discovered that the ANN algorithm was the most effective model for assessing solar radiation [20]. Alizamir et al. found that when predicting solar radiation, the GBT model outperforms other models in terms of accuracy and precision [21]. Srivastava et al. suggest that the four ML algorithms (CART, MARS, RF, and M5) can be utilized for forecasting hourly solar radiation for up to six days in advance, with RF demonstrating the best performance while CART showing the weakest performance [22]. In their study, Agbulut et al. demonstrated that the various ML algorithms they tested were highly accurate in predicting daily global solar radiation data, with the best performance achieved by the ANN algorithm [23].

Similar to solar energy, the prediction of wind energy poses a challenge due to its nonlinearity and randomness, which results in inconsistent power generation. Consequently, there is a need for an effective model to forecast wind energy, as evidenced by research studies [24,25]. In light of the rising global population and increasing energy demand, wind energy is viewed as a feasible alternative to depleting fossil fuels. Offshore wind farms are particularly advantageous compared to onshore wind farms since they offer higher capacity and access to more wind sources [26]. ML and DL models and algorithms are employed in wind energy development, utilizing wind speed data and other relevant information. Various researchers have proposed different models to increase prediction accuracy. For example, Zendehboud et al. suggested the SVM model as superior to other models and introduced hybrid SVM models [27]. Wang et al. proposed an HM comprising a combination of models for short-term wind speed prediction [28]. Demolli et al. used five ML algorithms to predict long-term wind power, finding that the SVR algorithm is most effective when the standard deviation is removed from the dataset [29]. Xiaoetal suggested using a self-adaptive kernel extreme learning machine (KELM) as a means to enhance the precision of forecasting [30]. The ARIMA and nonlinear autoregressive exogenous (NARX) models were evaluated by Cadenas et al., who concluded that the NARX model had less error [31]. Wind power and speed were predicted in other studies using a variety of models, including the improved dragonfly algorithm (IDA) with SVM (IDA–SVM) model, local mean decomposition (LMD), firefly algorithm (FA) models, and the CNN model [15,32,33].

ML and DL have significantly advanced the field of forecasting renewable energy. However, there are still several issues that need to be resolved. For instance, the choice of ML and DL algorithms, the selection of input data, and the handling of missing data are essential factors that affect the precision of forecasting models for renewable energy. Additionally, there is a need to develop robust and interpretable models that can provide insights into the factors that influence renewable energy generation.

This review provides an overview of the current advances and prospects of ML and DL algorithms for renewable energy forecasting. The paper highlights the advantages and limitations of different ML and DL algorithms and their applications to various renewable

energy sources. In addition, the paper discusses the challenges facing renewable energy forecasting with ML and DL and provides recommendations for future research in the field. This makes the article's approach distinctive since it acknowledges that addressing the variability and unpredictability of renewable energy sources is essential for achieving a more sustainable and dependable energy system. Additionally, it advocates the use of machine learning and deep learning models, a relatively new and creative strategy, as efficient instruments for tackling these problems. Utilizing these technologies will enable grid operators and renewable energy source operators to make better-educated decisions about managing and integrating renewable energy sources into the grid, resulting in a more efficient, reliable, and sustainable energy system.

## 2. Machine Learning-Based Forecasting of Renewable Energy

This section discusses the various machine learning techniques utilized for renewable energy forecasting. Machine learning-based forecasting has become an increasingly popular approach for predicting renewable energy output due to its ability to handle large and complex datasets. This section covers the two main categories of machine learning algorithms, supervised and unsupervised learning, and their various subcategories. It also explores reinforcement learning and its applications in renewable energy forecasting. The section provides a detailed description of each algorithm, along with its advantages, limitations, and applications in renewable energy forecasting.

### 2.1. Supervised Learning

ML is a subset of artificial intelligence that seeks to enable machines to learn from data and improve their ability to perform a particular task [34,35]. The process involves developing statistical models and algorithms that enable computers to identify patterns in data and utilize them to make decisions or predictions. In essence, ML involves teaching a computer to identify and react to specific types of data by presenting it with extensive examples, known as "training data." This training procedure helps the computer identify patterns and make predictions or decisions based on fresh data that it has not encountered previously [36–38]. The applications of ML span diverse industries such as healthcare, finance, e-commerce, and others [39–44]. In addition, ML techniques can be leveraged for predicting renewable energy generation, resulting in better management of renewable energy systems with improved efficiency and effectiveness. There are multiple ML algorithms available, each with distinct strengths and weaknesses. The algorithms can be categorized into three primary groups: supervised learning, unsupervised learning, and reinforcement learning [45].

Supervised learning refers to a ML method that involves training a model using data that has been labeled. The labeled data comprises input-output pairs, where the input is the data on which the model is trained and the output is the expected outcome [46,47]. The model learns to map inputs to outputs by reducing the error between the predicted and actual outputs during training. Once trained, the model can be applied to generate predictions on new, unlabeled data [48,49]. Regression and classification are the two basic sub-types of supervised learning algorithms (Figure 1) [46].

Table 1 presents a comparative analysis of various ML and DL algorithms, outlining their respective advantages and disadvantages across different applications.

1. Regression: Regression is a supervised learning approach that forecasts a continuous output variable based on one or more input variables. Regression aims to identify a mathematical function that can correlate the input variables to a continuous output variable, which may represent a single value or a range of values [50]. Linear regression, polynomial regression, and support vector regression (SVR) are the three main types of supervised learning algorithms in regression [51].

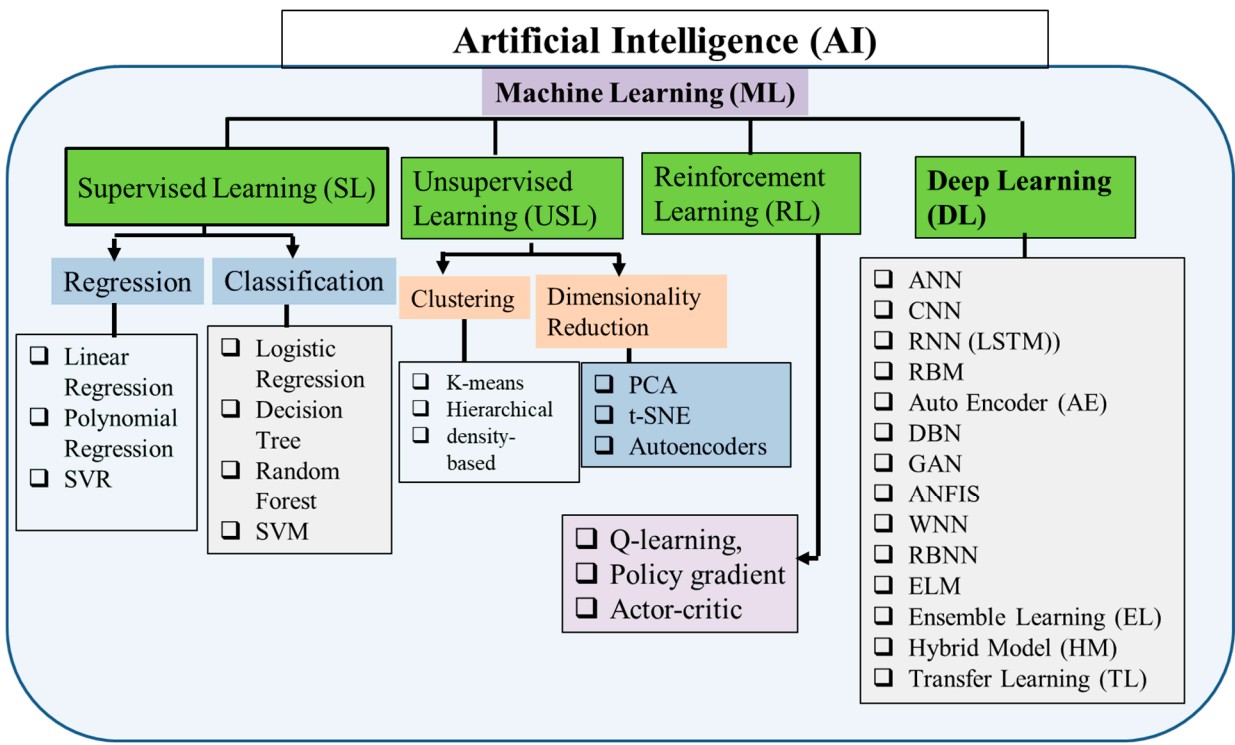

**Figure 1.** ML types and algorithms.

**Table 1.** ML and DL Technics with pros and cons in different applications.

| Technique | Pros | Cons | Applications |
|---|---|---|---|
| Linear Regression | Easy to implement, fast training | Limited to linear relationships | Predictive analytics |
| Logistic Regression | Interpretable, works well with small datasets | Assumes linearity, apply only for classification | Predict power outages, classify extreme weather events, market, and healthcare |
| Decision Trees | Interpretable, can handle both categorical and continuous data | Prone to overfitting | Predictive maintenance, finance |
| Random Forest | High accuracy, less prone to overfitting | Computationally expensive compared to DT, difficult to interpret | Operation control strategy, image classification, and fraud detection |
| SVM | Can handle high-dimensional data, can handle non-linear relationships, robust to noise | Computationally expensive and requires careful parameter tuning | Text classification, bioinformatics |
| K-means clustering | Simple and fast, useful for data exploration and segmentation | Requires a pre-determined number of clusters and can be sensitive to initial conditions | Market segmentation, image segmentation |
| PCA | Can reduce dimensionality and noise in data, useful for data exploration and visualization | May not capture all relevant information and can be difficult to interpret | Image and speech recognition, natural language processing |
| Reinforcement Learning | Can learn through trial and error, useful for decision-making in dynamic environments | Requires a lot of data and can be prone to overfitting | Game playing, robotics |
| ANN | Can learn complex relationships, handle large datasets, and model non-linear relationships | Requires large amounts of data and can be difficult to interpret | Predict energy demand (stationary), energy resource forecasting, image recognition, and speech recognition |
| CNN | Highly effective for image analysis, it can learn features automatically | Requires large amounts of data, is computationally expensive, may not be suitable for low spatial or temporal resolutions | Object detection, image classification, and predicting energy demand based on satellite images of areas |

**Table 1.** *Cont.*

| Technique | Pros | Cons | Applications |
|---|---|---|---|
| RNN | Can handle sequential data and time series data and can handle long-term dependencies | Can be prone to overfitting and slow training, and may suffer from vanishing or exploding gradients | Energy price forecasting (time series), speech recognition, and sentiment analysis |
| LSTM | Can handle long-term dependencies, which is useful for time series data | Can be prone to overfitting and require careful tuning | Time series, speech recognition, natural language processing, load forecasting, and energy price forecasting (time series) |
| Autoencoders | Can reduce dimensionality and noise in data and be used for unsupervised learning | Requires large amounts of data and can be difficult to interpret | Anomaly detection, image, and speech recognition |
| ELM | Fast training, can handle large datasets | Limited interpretability may not generalize well to new data | Renewable energy forecasting, image and speech recognition, predictive analytics |
| GRNN | Fast training and can handle noise in data | Limited to regression tasks and may not scale well to large datasets | Renewable energy forecasting, time series prediction, and function approximation |
| RBNN | Effective for non-linear regression and classification tasks | Requires careful tuning of network architecture and hyperparameters | Image and speech recognition, anomaly detection |
| WNN | Can handle multi-resolution and multi-scale data | Requires careful selection of wavelet basis functions and can be computationally expensive | Image and signal processing, time series prediction |
| ANFIS | Can handle uncertainty and non-linearity in data | Requires careful selection and tuning of fuzzy rules and can be computationally expensive | Control systems, fault diagnosis |
| DBN | Can learn hierarchical representations of data, which is effective for unsupervised learning | Requires large amounts of data, can be difficult to interpret | Image and speech recognition, natural language processing |
| Ensemble Learning | Can improve performance and reduce overfitting by combining multiple models | Can be computationally expensive and may require careful tuning | Renewable energy forecasting, image and speech recognition, and natural language processing |
| Transfer Learning | Can leverage pre-trained models to improve performance and require less data | May not generalize well to new data, limited to similar tasks | Load forecasting, energy price prediction, predictive maintenance, fault diagnosis, energy consumption, energy efficiency forecasting, renewable energy foresting, image and speech recognition, and natural language processing |

Linear and Polynomial Regression: Linear regression is a prevalent and straightforward approach used to forecast a continuous output variable utilizing one or more input variables. It uses a straight line to indicate the correlation between the input variables and the output variables [52]. On the other hand, polynomial regression, a type of linear regression, employs nth-degree polynomial functions to depict the connection between input features and the outcome variable [53]. This can enhance the accuracy of predictions by enabling the model to capture more intricate correlations between the input data and the target variable. In renewable energy forecasting, both linear and polynomial regression can be used to predict the power output of RES such as solar and wind power [54,55]. Weather information such as temperature, humidity, and wind speed are frequently included in the input characteristics, along with historical power output data. The target variable is the power output of the renewable energy source, which can be predicted using the input features.

For instance, Ibrahim et al. (2012) used data from a weather station collected over three years to create a linear regression model to predict solar radiation in Perlis. The model used three input variables (average daily maximum and minimum temperatures, as well as the average daily solar radiation) and had a good fit with an R-squared value of 0.954. The authors concluded that their model could be a useful tool for estimating

solar radiation in Perlis [56]. Ekanayake et al. (2021) created artificial neural networks (ANNs), multiple linear regression, and power regression models to produce wind power prediction models for a Sri Lankan wind farm. In their modeling approach, they utilized climate parameters such as average wind speed and average ambient temperature as input variables for their analysis. The models were developed using power generation data over five years and showed acceptable accuracy with low RMSE, low bias, and a high correlation coefficient. The ANN model was the most precise, but the MLR and PR models provide insights for additional wind farms in the same area [57]. Mustafa et al. (2022) also compared four regression models, linear regression, logistic regression, Lasso regression, and elastic regression, for solar power prediction. The results showed that all four models are effective, but the elastic regression outperformed the others in predicting maximum solar power output. Principal component analysis (PCA) was also applied, showing improved results in the elastic regression model. The paper focuses on the strengths and weaknesses of each solar power prediction model [58].

Support Vector Regression (SVR): The SVR algorithm is utilized in regression analysis within the field of ML [59]. It works by finding the best hyperplane that can separate the data points in a high-dimensional space. The selection of the hyperplane aims to maximize the distance between the closest data points on each side of it. The approach involves constraining the margin while minimizing the discrepancy between the predicted and actual values. It is also a powerful model to predict renewable energy potential for a specific location. For example, Yuan et al. (2022) proposed a jellyfish search algorithm optimization SVR (IJS-SVR) model to predict wind power output and address grid connection and power dispatching issues. The SVR was optimized using the IJS technique, and the model was tested in both the spring and winter. IJS-SVR outperformed other models in both seasons, providing an effective and economical method for wind power prediction [60]. In addition, Li et al. (2022) created ML-based algorithms for short-term solar irradiance prediction, incorporating a hidden Markov model and SVM regression techniques. The Bureau of Meteorology demonstrated that their algorithms can effectively forecast solar irradiance for 5–30 min intervals in various weather conditions [61]. Mwende et al. (2022) developed SVR and random forest regression (RFR) models for real-time photovoltaic (PV) power output forecasting. On the validation dataset, SVR performed better than RFR with an RMSE of 43.16, an adjusted $R^2$ of 0.97, and a MAE of 32.57, in contrast to RFR's RMSE of 86, an adjusted $R^2$ of 0.90, and a MAE of 69 [62].

2. Classification: Classification, a form of supervised learning, involves using one or more input variables to anticipate a categorical output variable [63]. Classification aims to find a function that can map the input variables to discrete output categories. The most widely used classification algorithms for predicting RES include logistic regression, decision trees, random forests, and support vector machines.

Logistic Regression: Logistic regression is a classification method that utilizes one or more input variables to forecast a binary output variable [64,65]. It models the probability of the output variable being true or false using a sigmoid function. In renewable energy forecasting, logistic regression can be used to predict whether or not a specific event will occur, such as a solar or wind farm reaching a certain level of power output. For instance, Jagadeesh et al. used ML to develop a forecasting method for solar power output in 2020. They used a logistic regression model with data from 11 months, including plant output, solar radiation, and local temperature. The study found that selecting the appropriate solar variables is essential for precise forecasting. Additionally, it examined the algorithm's precision and the likelihood of a facility generating electricity on a particular day in the future [66].

Decision Trees: An alternative classification method is decision trees, which involve dividing the input space into smaller sections based on input variable values and then assigning a label or value to each of these sections [65]. The different studies developed decision tree models to forecast power output from different renewable energy systems. Essama et al. (2018) developed a model to predict the power output of a photovoltaic (PV)

system in Cocoa, Florida, USA, using weather parameters obtained from the United States National Renewable Energy Laboratory (NREL). By selecting the best performance among the ANN, RF, DT, extreme gradient boosting (XGB), and LSTM algorithms, they aim to fill a research gap in the area. They have concluded that even if all of the algorithms were good, ANN is the most accurate method for forecasting PV solar power generation.

Random Forest: An effective and reliable prediction is produced by the supervised ML method known as random forest, which creates several decision trees and merges them [67]. The bagging technique, which is employed by random forests, reduces the variance of the base algorithms. This technique is particularly useful for forecasting time-series data [68]. Random forest mitigation correlation between trees by introducing randomization in two ways: sampling from the training set and selecting from the feature subset. The RF model creates a complete binary tree for each of the N trees in isolation, which enables parallel processing.

Vassallo et al. (2020) investigate optimal strategies for random forest (RF) modeling in wind speed/power forecasting. The investigation examines the utilization of random forest (RF) as a corrective measure, comparing direct versus recursive multi-step prediction, and assessing the impact of training data availability. Findings indicate that RF is more efficient when deployed as an error-correction tool for the persistence approach and that the direct forecasting strategy performs slightly better than the recursive strategy. Increased data availability continually improves forecasting accuracy [69]. In addition, Shi et al. (2018) put forward a two-stage feature selection process, coupled with a supervised random forest model, to address overfitting, weak reasoning, and generalization in neural network models when forecasting short-term wind power. The proposed methodology removes redundant features, selects relevant samples, and evaluates the performance of each decision tree. To address the inadequacies of the internal validation index, a new external validation index correlated with wind speed is introduced. Simulation examples and case studies demonstrate the model's better performance than other models in accuracy, efficiency, and robustness, especially for noisy data and wind power curtailment [70]. Similarly, Natarajan and Kumar (2015) also compared wind power forecasting methods. Physical methods rely on meteorological data and numerical weather prediction (NWP), while statistical methods such as ANN and SVM depend on historical wind speed data. This study experiments with the random forest algorithm, finding it more accurate than ANN for predicting wind power at wind farms [71].

Support Vector Machines (SVM): SVM are a type of classification algorithm that identifies a hyperplane and maximizes the margin between the hyperplane and the data points, akin to SVR [72,73]. SVM has been utilized in renewable energy forecasting to estimate the power output of wind and solar farms by incorporating input features such as historical power output, weather data, and time of day. For instance, Zeng et al. (2022) propose a 2D least-squares SVM (LS-SVM) model for short-term solar power prediction. The model uses atmospheric transmissivity and meteorological variables and outperforms the reference autoregressive model and radial basis function neural network model in terms of prediction accuracy [74]. R. Meenal and A. I. Selvakumar (2018) conducted studies comparing the accuracy of SVM, ANN, and empirical solar radiation models in forecasting monthly mean daily global solar radiation (GSR) in several Indian cities using varying input parameters. Using WEKA software, the authors determine the most significant parameters and conclude that the SVM model with the most influential input parameters yields superior performance compared to the other models [75]. Generally, classification algorithms are used to predict categorical output variables, and regression techniques are used to predict continuous output variables. The particular task at hand and the properties of the data will determine which method is used.

### 2.2. Unsupervised Learning

Another form of ML is unsupervised learning, where an algorithm is trained on an unlabeled dataset lacking known output variables, to uncover patterns, structures, or

relationships within the data [76–78]. Unsupervised learning algorithms can be primarily classified into two types, namely clustering and dimensionality reduction [79].

Clustering: It is an unsupervised learning method that consists of clustering related data points depending on how close or similar they are to one another. Clustering algorithms, such as K-means clustering, hierarchical clustering, and density-based clustering, are commonly used in energy systems to identify natural groupings or clusters within the data. The primary objective of clustering is to discover these inherent patterns, or clusters [76,77]. K-means clustering is a widely used approach for dividing data into k clusters, where k is a user-defined number. Each data point is assigned to the nearest cluster centroid by the algorithm, and the centroids are updated over time using the average of the data points in the cluster [76,77]. Hierarchical clustering is also a family of algorithms that recursively merge or split clusters based on their similarity or distance to create a hierarchical tree-like structure of clusters. The other family of clustering algorithms that groups data points that are within a certain density threshold and separates them from areas with lower densities is the density-based clustering algorithm [76–78].

Dimensionality Reduction: It is also an unsupervised learning technique utilized to decrease the quantity of input variables or features while retaining the significant information or structure in the data [76–78]. The purpose of dimensionality reduction is to find a lower-dimensional representation of data that captures the majority of the variation or variance in the data. Principal component analysis (PCA), t-SNE, and autoencoders are some dimensionality reduction algorithms used in renewable energy forecasting [79]. Principal component analysis (PCA) is a commonly utilized method for decreasing the dimensionality of a dataset. It does so by identifying the primary components or directions that have the most variability in the data and then mapping the data onto these components [79]. t-SNE is a non-linear dimensionality reduction algorithm that is particularly useful for visualizing high-dimensional data in low-dimensional space. It uses a probabilistic approach to map similar data points to nearby points in low-dimensional space. Autoencoders are a type of neural network that can learn to encode and decode high-dimensional data in a lower-dimensional space. The encoder network is trained to condense the input data into a representation with fewer dimensions, and the decoder network is trained to reconstruct the original data from this condensed representation [79].

In general, unsupervised learning algorithms are particularly useful when there is a large amount of unstructured data that needs to be analyzed and when it is not clear what the specific target variable should be. Unsupervised learning has found various applications in the field of renewable energy forecasting, and one of its commonly used applications is the clustering of meteorological data [80]. For example, in a study by J. Varanasi and M. Tripathi (2019), K-means clustering was used to group days of the year, sunny days, cloudy days, and rainy days into clusters based on similarity for short-term PV power generation forecasting [81]. The resulting clusters were then used to train separate ML models for each cluster, which resulted in improved PV power forecasting accuracy. Unsupervised learning has also been used for anomaly detection in renewable energy forecasting. Anomaly detection refers to the task of pinpointing data points that exhibit notable deviations from the remaining dataset. In the context of renewable energy forecasting, anomaly detection can aid in identifying exceptional weather patterns or uncommon circumstances that may impact renewable energy generation. For example, in a study by Xu et al. (2015), the K-means algorithm was used to identify anomalous wind power output data, which was then employed to improve the accuracy of the wind power forecasting model [82].

In the realm of renewable energy forecasting, unsupervised learning has been utilized for feature selection, which involves choosing a smaller set of pertinent features from a larger set of input variables. In renewable energy forecasting, feature selection can be used to reduce the computational complexity of ML models and improve the accuracy of renewable energy output predictions. For example, in a study by Scolari et al. (2015),

K-means clustering was used to identify a representative subset of features for predicting solar power output [83].

Overall, unsupervised learning is a powerful tool for analyzing large amounts of unstructured data in renewable energy forecasting. Clustering, anomaly detection, and feature selection are just a few of the many applications of unsupervised learning in this field, and new techniques are continually being developed to address the unique challenges of renewable energy forecasting.

### 2.3. Reinforcement Learning Algorithms

Reinforcement learning (RL) is a branch of ML in which an agent learns to make decisions in an environment to maximize a cumulative reward signal [84,85]. The agent interacts with its surroundings by taking actions and receiving responses in the form of rewards or penalties that are contingent on its actions. [86]. Some examples of RL algorithms are Q-learning, policy gradient, and actor-critic [87,88]. Q-learning is a RL algorithm used for learning optimal policies for decision-making tasks by iteratively updating the Q-values, which represent the expected future rewards for each action in each state [89]. Policy gradient is also a RL algorithm used for learning policies directly without computing the Q-values [90]. Actor-critic is another RL algorithm that combines elements of both value-based and policy-based methods by training an actor network to generate actions and a critic network to estimate the value of those actions [90].

Renewable energy forecasting is among the many tasks for which RL has been utilized [88,91]. One approach to applying RL to renewable energy forecasting is to use it to control the operation of energy systems [92]. For example, Sierra-García J. and S. Matilde (2020) developed an advanced yaw control strategy for wind turbines based on RL [93]. This approach uses a particle swarm optimization (PSO) and Pareto optimal front (PoF)-based algorithm to find optimal actions that balance power gain and mechanical loads, while the RL algorithm maximizes power generation and minimizes mechanical loads using an ANN. The strategy was validated with real wind data from Salt Lake City, Utah, and the NREL 5-MW reference wind turbine through FAST simulations [93].

### 2.4. Deep Learning (DL)

DL is a type of ML that employs ANNs containing numerous layers to acquire intricate data representations with multiple layers of abstraction. The term "deep" refers to the large number of layers in these ANNs, which can range from a few layers to hundreds or even thousands of layers [94]. DL algorithms can learn to recognize patterns and relationships in data through a process known as "training." During training, the weights of the links between neurons in an ANN are changed to reduce the disparity between the anticipated and actual output [95]. DL has brought about significant transformations in several domains, such as energy systems, computer vision, natural language processing, speech recognition, and autonomous systems. It has facilitated remarkable advancements in various fields, such as natural language processing, game playing, speech recognition, and image recognition [80].

## 3. DL Algorithms Used for Renewable Energy Forecasting

### 3.1. ANN for Renewable Energy Forecasting

Artificial neural networks (ANNs) belong to a category of ML models that imitate the arrangement and operation of the human brain [96]. They are designed to learn from data and utilize that knowledge to produce predictions or decisions. At a high level, ANNs consist of three main components: input layers, hidden layers, and output layers. The input layer receives the data, which is usually represented as a vector of numbers. The output layer produces the desired output of the network, which could be a classification (e.g., predicting the type of an object in an image) or a regression (e.g., predicting the price of a house based on its features). The hidden layers are where most of the "computation" happens in the network. They consist of one or more layers of neurons that perform

nonlinear transformations on the input data [97]. Each neuron in an ANN receives input from other neurons or directly from the input layer. For each input, a weight is assigned that signifies the connection's potency between two neurons (Figure 2). Then, the neuron processes an activation function on the weighted sum of its inputs, which generates an output. This output can serve as the input for another neuron, and this process repeats until the final output of the output layer is obtained [98].

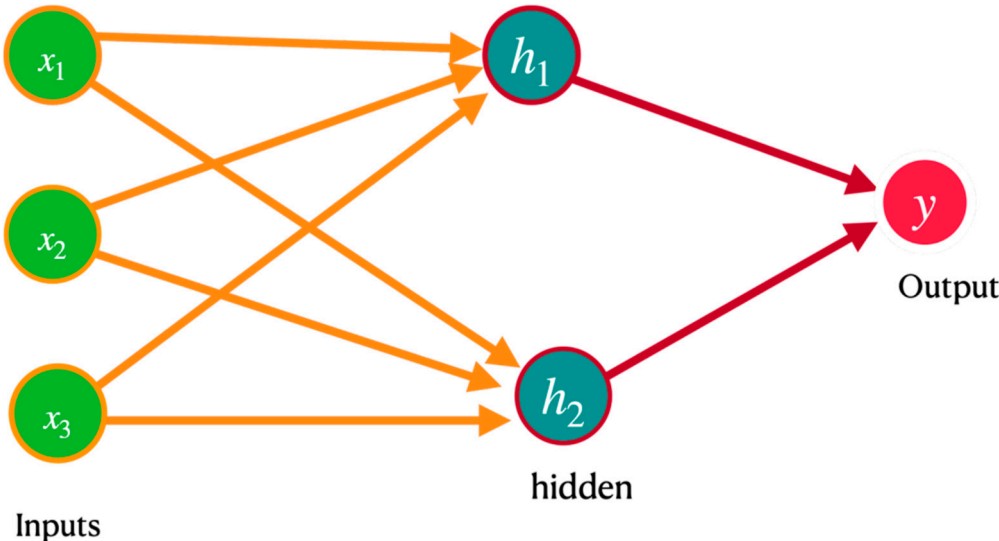

**Figure 2.** Artificial neural network architecture.

　　The weights within the network are modified during training to ensure that the network generates the intended output for a given input. This process is usually executed through a technique known as back-propagation, which determines the gradient of the loss function concerning the weights and adjusts them correspondingly (Figure 2). [99,100]. The loss function quantifies the variance between the predicted output and the actual output, and the primary objective of the training is to lessen this discrepancy. ANNs are versatile and can be employed for various purposes, such as predicting renewable energy, classifying images, recognizing speech, processing natural language, and conducting predictive analytics [101–105].

　　ANNs have been employed in renewable energy forecasting, such as solar energy, wind energy, and multi-renewable energy forecasting, for several years, demonstrating their efficacy in this application. For instance, Rehman and Mohandes [106] used ANN to estimate global solar radiation in Abha, Saudi Arabia, by incorporating air temperature and relative humidity as inputs. Meanwhile, Benghanem et al. [107] established six ANN-based models to estimate global solar radiation in Al-Madinah, Saudi Arabia, and found that the model based on sunshine duration and air temperature had the highest precision. In a different study, Ozgoren et al. [108] developed an ANN model that predicted monthly global solar radiation in Turkey by using several input variables and the stepwise MNLR technique to identify influential inputs. The ANN model demonstrated an acceptable level of precision compared to measured values. Despite the variations in the models and input variables used, ANN has proven to be a reliable tool for predicting global solar radiation in different regions. S. Kumar and T. Kaur (2016) also used ANN to predict solar radiation for solar energy applications in Himachal Pradesh. The ANN model used temperature, rainfall, sunshine hours, humidity, and barometric pressure as input variables. Three models with 3 to 5 input parameters were developed and tested, with the ANN-I5 model showing the best prediction accuracy with a mean absolute percentage error (MAPE) of 16.45%. This study showed the method can also be used to identify solar energy potential for any location worldwide without direct measuring instruments [109]. Another study by N. Premalathaa and Amirtham V. (2016) showed that an ANN model for accurately

predicting solar radiation using meteorological mean monthly data such as station level pressure, ambient air temperature, wind speed, average GSR, and relative humidity from five different locations across India over the last 10 years. The analysis evaluated two ANN models that incorporated four distinct algorithms, and the optimal algorithm and model were determined by the minimum MAE and RMSE, as well as the maximum R. The resulting ANN model exhibits a low MAPE, which renders it useful for designing or assessing solar energy systems in regions of India that lack meteorological data recording capabilities [110]. In their study, Woldegiyorgis et al. (2021) explored the viability of utilizing ANN for estimating the mean daily global solar radiation (GSR) and contrasted its effectiveness with empirical models founded on sunshine. They used daily average sunshine hours, temperature (max/min), wind speed, relative humidity, pressure, and the number of days as inputs, with daily averaged global solar radiation as the network output. The results demonstrated that the ANN model displayed favorable performance with a validation R-value of 0.932, surpassing the empirical models [111].

Wind energy forecasting is another important area of research in renewable energy forecasting. For instance, Jamii et al. (2022) proposed an ANN-based paradigm to forecast wind power generation and load demand using meteorological parameters such as wind speed, atmospheric pressure, and temperature as inputs. Results showed that the ANN outperformed four other ML methods, providing high effectiveness and accuracy for power forecasting [112]. Q. Chen and K. Folly (2019) suggested an artificial neural network (ANN) model for precise short-term wind power prediction in small wind farms. Their research examines how the input variables (wind speed at various heights, wind direction, atmospheric pressure, temperature, and relative humidity) and sample size influence the forecasting efficiency and computational expense of the model. The study investigates the effect of input variables and sample size on the forecasting performance and computing cost of the model. Their findings suggest that the ANN model with all input features and a large training sample size performs the best in terms of forecasting [113].

*3.2. CNN for Renewable Energy Forecasting*

Convolutional neural networks (CNNs) are a class of ANN that exhibit exceptional performance in handling and interpreting data with a grid-like arrangement, such as videos or images [114]. CNNs derive inspiration from the organization and operation of the visual cortex in the brain and employ convolutional layers to acquire localized characteristics from the input data [115]. A CNN is composed of several layers, which encompass convolutional layers, pooling layers, and fully connected layers (Figure 3). The convolutional layers are the principal building blocks of a CNN, which utilize filters (referred to as kernels) to extract localized characteristics from the input data [116]. Each filter is a small matrix that slides over the input data, performing a dot product between the filter and the corresponding input pixels. The output of this operation is a feature map, which highlights the areas of the input data that are most important for the task at hand [116,117]. Pooling layers are frequently added after convolutional layers to decrease the spatial dimensions of feature maps and enhance the computational efficiency of the network. Popular types of pooling include max pooling, which chooses the highest value within a small area of the feature map, and average pooling, which calculates the mean value within a small area of the feature map [118]. The fully connected layers are employed to generate the ultimate predictions based on the features acquired by the convolutional and pooling layers. These layers are similar to those in a traditional neural network and use the learned features to make predictions based on the task at hand. In the course of training, the CNN weights are modified to minimize the disparity between the predicted and actual output, akin to other types of neural network structures [119]. The back-propagation algorithm is utilized to compute the gradients of the loss function relative to the weights, and the weights are subsequently updated based on this information [120].

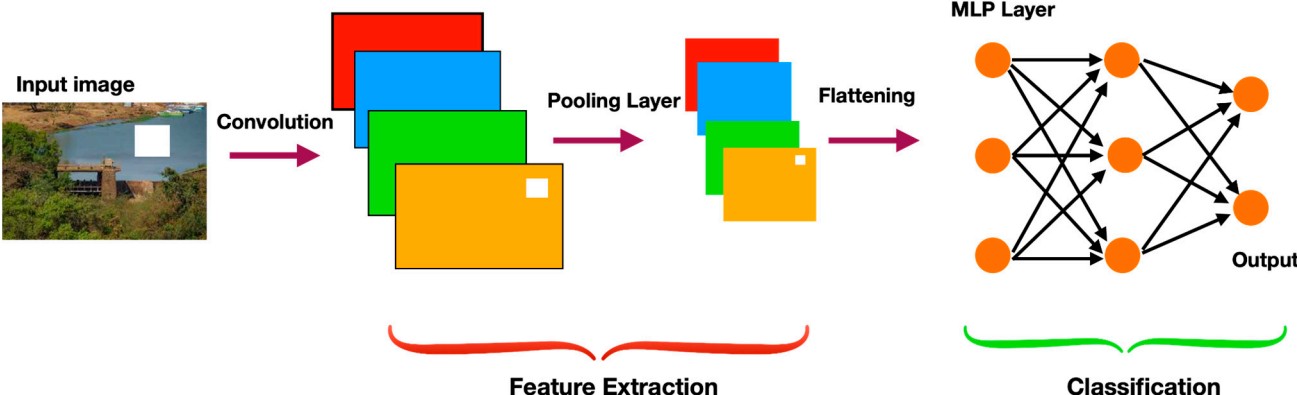

**Figure 3.** Convolutional neural network architecture [120].

CNNs have demonstrated exceptional performance in various image and video recognition tasks, such as object detection, segmentation, and classification. To forecast time series, which is essential for forecasting renewable energy, CNNS can also be used in combination with the other algorithms [121–124]. For example, Lim et al. (2022) propose a CNN-LSTM for stable power generation forecasting in photovoltaic (PV) systems, considering the impact of environmental factors such as solar radiation and temperature. PV power output data from a plant located in Busan, Korea, was used to train and test the model. The suggested model yielded a mean absolute percentage error of 4.58 on a clear day and 7.06 on a cloudy day, demonstrating its ability to enhance the efficiency of PV power plant operations [125]. Similarly, Gao et al. (2019) propose a CEEMDAN–CNN–LSTM model for hourly solar irradiance forecasting to managing electricity generation and smart grids. The model employs CEEMDAN to decompose data into constitutive series and a DL network based on CNN and LSTM to predict solar irradiance. The model outperforms other methods, achieving an average RMSE of 38.49 W/m$^2$ and demonstrating stable performance in different climates [126]. Cannizzaro et al. (2021) presented a fresh approach to anticipating short- and long-term global horizontal solar irradiance (GHI) through machine learning techniques. Their methodology involves a combination of variational mode decomposition (VMD) and CNN with either RF or LSTM. The approach is evaluated on a real-world dataset and achieves accurate results [127]. Furthermore, Wu et al. (2020) suggest a spatio-temporal correlation model (STCM) that utilizes CNN-LSTM to forecast ultra-short-term wind power. The model reconstructs meteorological factors at different sites from input data using CNN to extract spatial correlation features and LSTM to extract temporal correlation features. The STCM performs better than traditional models and accurately forecasts wind power using measured meteorological factors and wind power datasets from a wind farm in China [128].

*3.3. RNN for Renewable Energy Forecasting*

Recurrent neural networks (RNNs) are an artificial neural network category that is specifically engineered to manage sequential data by analyzing every element in a sequence while retaining an internal state or memory of previous elements [129–131]. RNNs are particularly useful for natural language processing, speech recognition, time series prediction, and other applications that involve sequential or temporal data. The key feature of RNNs is the use of recurrent connections, which allow information to be passed from one-time step to the next. Recurrent connections are established in the network through the inclusion of loops, which permit the output of the prior time step to be utilized as input for the current time step. Similar to other neural networks, an RNN consists of layers of neurons with learnable weights. However, unlike other neural networks, the input and output of an RNN can be sequences of variable length, and the weights are shared across all time steps, allowing the network to learn patterns that are dependent on the sequence of input data (Figure 4) [132]. The internal state of an RNN at each time step is typically

represented by a hidden vector or memory cell. The hidden vector is updated at each time step by combining the input at that time step with the previously hidden vector using a set of learnable weights. This update is usually performed using an activation function, such as the hyperbolic tangent or the rectified linear unit (ReLU). One limitation of basic RNNs is that they can struggle to capture long-term dependencies in the input sequence, which can cause the gradient to vanish or explode during training. Recurrent connections are established in the network through the inclusion of loops, which permit the output of the prior time step to be utilized as input for the current time step. Different types of RNNs have been created to tackle this issue, including long short-term memory (LSTM) networks and gated recurrent units (GRUs), which employ more intricate structures to effectively retain long-term correlations in the input sequence [133]. During the training process, the RNN weights are modified to reduce the difference between the predicted output and the actual output, similar to other neural network models. The back-propagation algorithm is employed to determine the gradients of the loss function in relation to the weights, and the weights are subsequently updated based on this information [134].

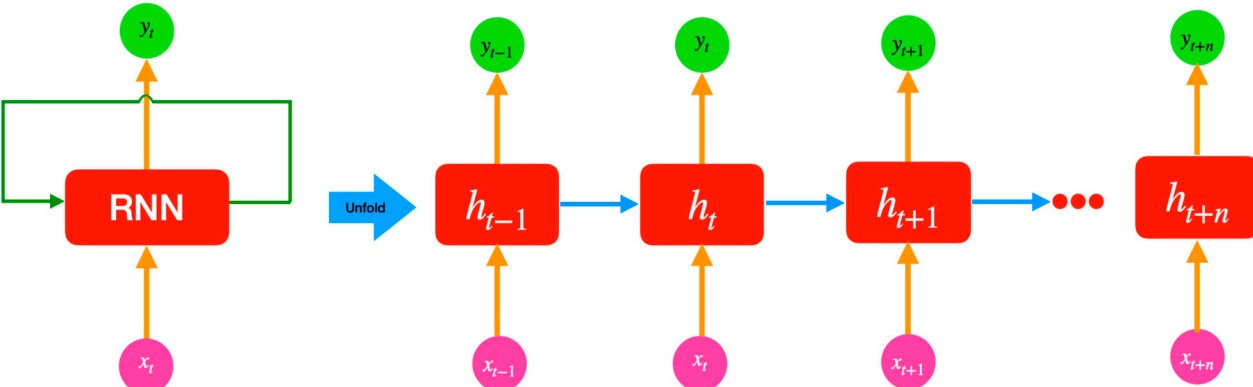

**Figure 4.** Architecture of RNN.

RNNs have shown remarkable success in tasks involving sequential data, such as speech recognition, sentiment analysis, machine translation, and even time series forecasting, including renewable energy forecasting. Several research studies have investigated the use of RNNs for predicting renewable energy. One such study by Kisvari et al. (2021) suggests a data-driven method for wind power prediction that involves pre-processing, anomaly detection, feature engineering, and hyperparameter tuning using gated recurrent DL models [135]. They also compare a new DL neural network of GRU with LSTM. The approach achieves high accuracy at lower computational costs, and GRU outperforms LSTM in predictive accuracy [135]. Another work by M. Abdel-Nasser and K. Mahmoud (2017) proposes using LSTM-RNN to precisely estimate PV power output. Because of their recurrent architecture and memory units, LSTM networks can simulate temporal variations in PV output power. Utilizing hourly datasets, the proposed strategy is evaluated and found to reduce forecasting errors more than three previous methods [136]. Yadav et al. (2013) also suggest a RNN model that utilizes an adaptive learning rate to predict daily, mean monthly, and hourly solar irradiation with the assistance of meteorological data. The outcomes demonstrate that the RNN performs better than the multi-layer perceptron (MLP) method, and the proposed adaptive learning rate enhances performance in comparison to the traditional feed-forward network [137].

### 3.4. RBM for Renewable Energy Forecasting

Restricted Boltzmann Machines (RBM) is a type of unsupervised neural network that can learn complex probability distributions over input data. They are composed of two layers, a hidden layer and a visible layer, with each layer consisting of binary nodes that are either activated or deactivated [138]. The training process of RBM involves contrastive

divergence, a technique that works towards reducing the dissimilarity between the input data and the model's depiction of the data. Through training, the RBM adapts the weights connecting the visible and hidden layers to model the probability distribution of the input data. RBMs have several unique features that make them useful for a variety of applications. Their capacity to learn high-level representations of input data without labels or supervision is one of their key strengths. This makes them ideal for unsupervised learning tasks like feature learning and dimensionality reduction [139]. Another strength of RBMs is their ability to model complex dependencies between input features, which makes them effective in modeling data with multiple interacting factors. They have been effectively employed in a wide range of fields, including image recognition, voice recognition, and natural language processing. Finally, RBMs have also been used as building blocks for more complex neural networks, such as deep belief networks and deep neural networks. In these architectures, RBMs are used to pre-train the network's layers before fine-tuning them for a specific task (Figure 5) [140].

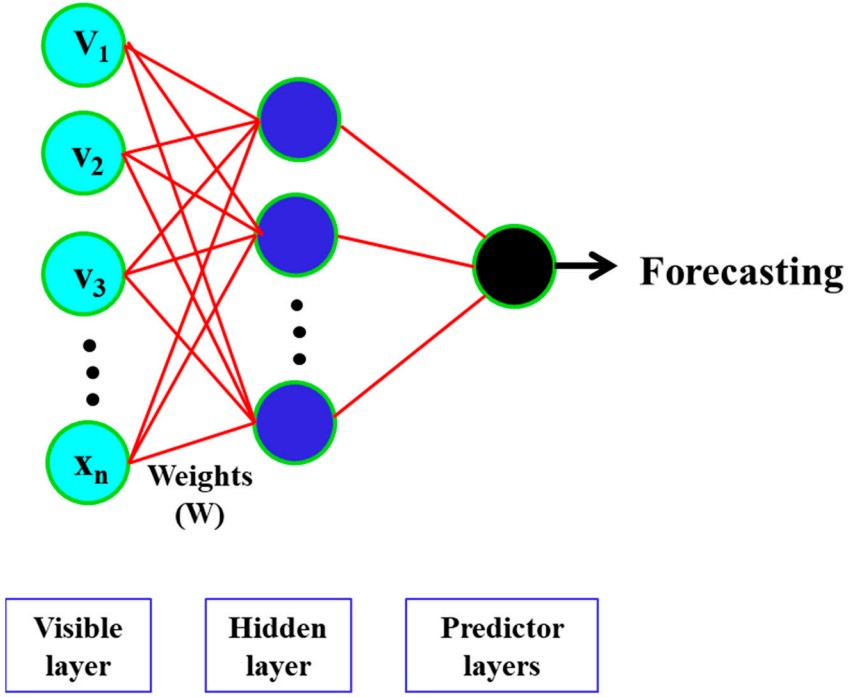

**Figure 5.** RBM forecasting architecture [138].

RBMs have found applications in various fields, including renewable energy forecasting. One such application involves using RBMs as a pre-processing step to extract features from renewable energy data before using them in other ML models such as neural networks for forecasting. Yang et al., for instance, proposed an unsupervised model for identifying irregularities in wind turbine monitoring systems that includes RBM [141].

*3.5. Auto Encoder for Renewable Energy Forecasting*

One of the most effective unsupervised learning models in recent decades is the autoencoder based on a deep neural network. The unsupervised model allows for the extraction of effective and discriminative features from a large unlabeled data set, making this approach extensively suitable for feature extraction and dimensionality reduction [142]. Essentially, an autoencoder can be described as a neural network consisting of three fully connected layers, with the encoder containing input and hidden layers and the decoder containing hidden and output layers. The encoder converts higher-dimensional input data into a lower-dimensional feature vector [143]. The data is then converted back to the input dimension by the decoder. Building a complex nonlinear relationship between the input

data and the output data is one of the deep neural network's top priorities since it enables the autoencoder to successfully recreate the decoder's output. As a result, throughout the entire training period, the reconstruction error will decrease simultaneously, and important features will be stored in the hidden layer. Lastly, the output of the hidden layer will show how effectively the proposed autoencoder extracted features [144,145]. The basic autoencoder's configuration is shown in Figure 6.

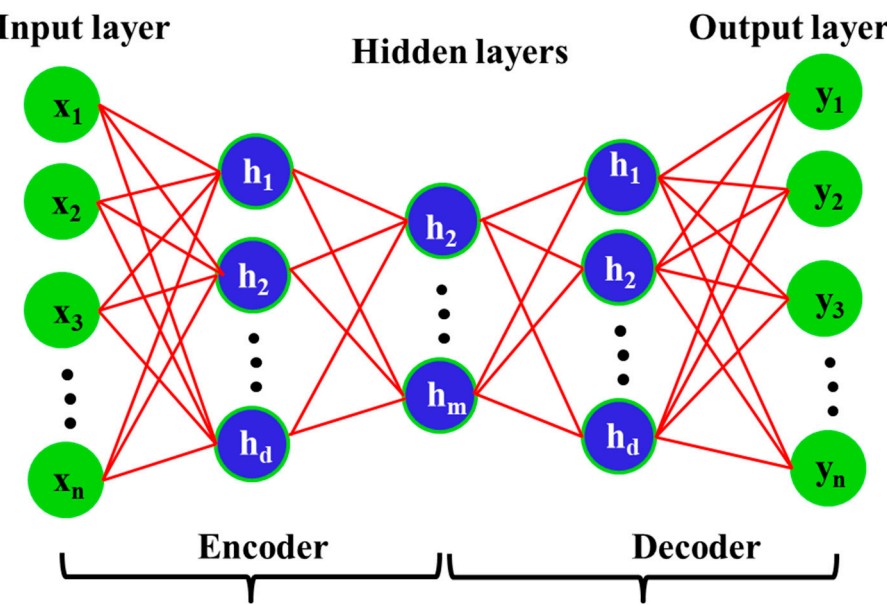

**Figure 6.** The autoencoder's basic design [145].

In renewable energy forecasting, autoencoder models have been used to extract features from input data such as weather data, historical energy production data, and other relevant variables. These features are then used to train ML models for energy forecasting. For example, Dairi et al. (2015) propose a variational autoencoder (VAE) model for short-term solar power forecasting. The study compares the performance of the VAE-based method with seven DL methods and two ML methods using data from two PV systems. Results indicate that the VAE consistently outperforms the other methods in forecasting accuracy, highlighting the superiority of DL techniques over traditional ML methods [146]. Jaseena and Kovoor (2015) also presented a wind speed forecasting model that utilizes a hybrid approach of an autoencoder and LSTM. The model incorporates an autoencoder to extract characteristics from the input data. The extracted features are then fed into an LSTM model for forecasting wind speed. Compared to other models, the proposed model attained superior accuracy in its forecasting [147].

### 3.6. Deep Belief Neural Networks (DBN) for Renewable Energy Forecasting

DBNs are deep neural networks made up of several layers of RBMs. Similar to RBMs, DBNs are generative models trained through unsupervised learning techniques to extract and represent features from input data [139]. The training process of DBNs follows a layer-wise unsupervised learning approach, where each layer is trained independently to extract features from input data (Figure 7) [148]. After training each layer, the output of the preceding layer is used as input to the subsequent layer until the entire network is trained. Once the network is fully trained, it can be fine-tuned using supervised learning methods for classification or regression tasks [149].

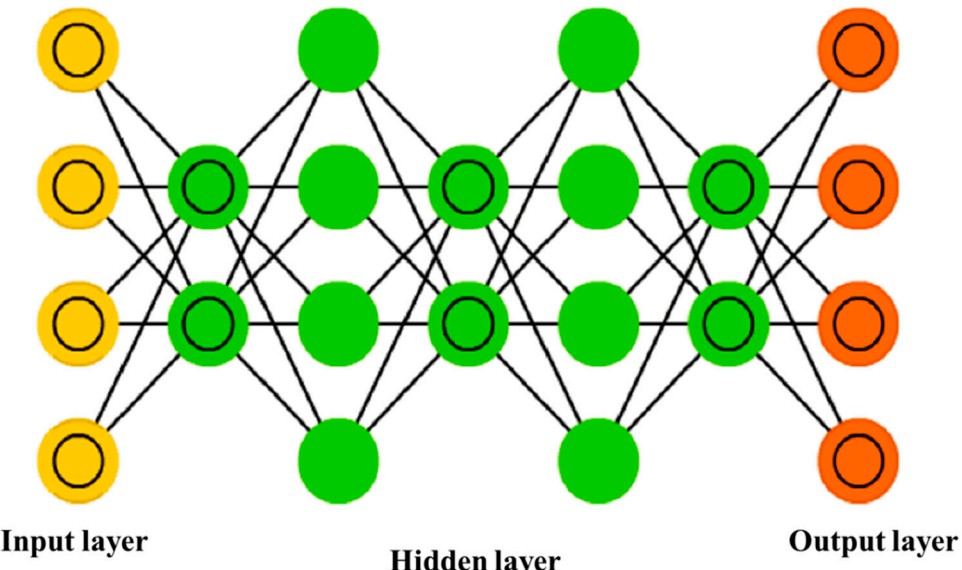

**Figure 7.** Architecture of DBN [138].

DBNs have found diverse applications in fields such as computer vision, speech recognition, and natural language processing, demonstrating remarkable performance in numerous use cases [150]. For example, DBNs have been utilized for image classification, achieving state-of-the-art accuracy on the MNIST dataset, which is a collection of handwritten digits. In natural language processing, DBNs have been used for sentiment analysis and language modeling, where they are effective in learning hierarchical representations of the input data. In the context of renewable energy forecasting, DBNs have been used for feature extraction and forecasting. For instance, a study by Noe et al. (2017) proposed a forecasting method based on DL and the deep belief network training algorithm. Their study found that the optimal number of input variables influences performance, and the proposed DBN accurately selected the parameters that best fit the data to achieve the lowest prediction error. The results were validated by comparing them to ELM, and actual evidence shows that DBN has better forecasting accuracy [151]. Similarly, Wang et al. (2018) introduced a deep belief network (DBN) model that employs clustered numerical weather prediction (NWP) data to enhance wind power forecasting, outperforming back-propagation neural networks (BP) and Morlet wavelet neural networks (MWNN) by over 44% in accuracy when tested on the Sotavento wind farm [152].

*3.7. ANFIS for Renewable Energy Forecasting*

ANFIS is a hybrid of ANN and fuzzy logic, designed in the early 1990s [153]. It uses a fuzzy inference system to approximate nonlinear functions, making it a powerful general estimator [154]. ANFIS consists of five layers, which include the input layer, fuzzification layer, rule layer, normalization layer, and output layer. The fuzzification layer is the first layer and takes in input values to determine membership functions using the premise parameter sets a, b, and c. The second layer, the rule layer, generates firing strengths for the rules. The third layer normalizes the firing strengths by dividing each value by the total firing strength. In the fourth layer, the normalized values and result parameter sets p, q, and r are used as inputs to produce defuzzified values, which are then sent to the final layer to generate the ultimate output [155]. Figure 8 illustrates the ANFIS architecture.

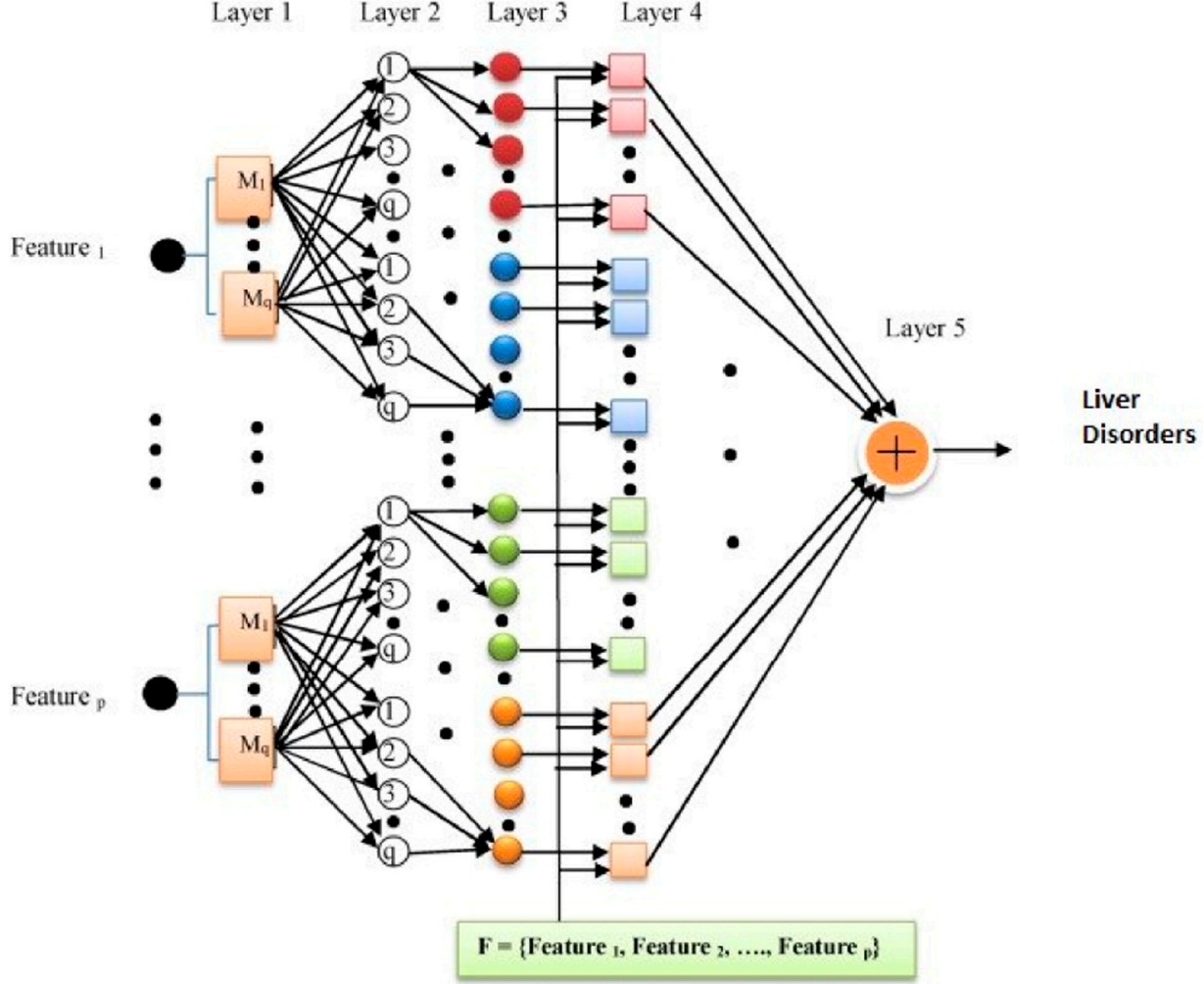

**Figure 8.** ANFIS architecture [156].

ANFIS has been extensively utilized in renewable energy forecasting owing to its capacity to capture both linear and nonlinear connections in the data. For instance, Hussieny et al. (2018) developed a wind speed and temperature prediction system utilizing a combination of ANN, a genetic algorithm fused with ANN (GPANN), and ANFIS. The implementation of ANFIS with a trapezoidal membership function yielded the best results, with an optimal mean square error of 7.2989 m/s for wind speed and 3.8364 °C for temperature [157]. Mellit et al. (2007) also proposed a new approach for estimating total solar radiation data with an ANFIS that is purely dependent on the mean sunshine duration and air temperature. The model was trained and tested using a 10-year database of daily sunshine duration, ambient temperature, and total solar radiation data. The validation data set produced a highly accurate estimate, with a mean relative error (REM) of less than 1% between the actual and predicted values and a correlation coefficient of 98%. They further claim that the proposed approach can be applied to any geographical location on Earth [158]. To enhance the accuracy of wind speed forecasting, ANFIS models have been utilized in conjunction with other machine learning models. Ahmed et al. (2017) conducted a study in which they developed a hybrid algorithm for wind speed forecasting, a critical factor for wind power generation, by introducing a novel optimization algorithm called Krill Herd (KH) and combining it with ANFIS. The Krill-ANFIS model, which was optimized using KH, outperformed the basic ANFIS, PSO-ANFIS, and GA-ANFIS models in terms of performance measures [157]. Yadav et al. (2019) also developed a hybrid model using a genetic algorithm and an adaptive network-based fuzzy inference system to forecast short-term

solar photovoltaic (PV) power in the Indian electricity market. This model outperformed four other models, demonstrating higher accuracy in PV power forecasting [159].

### 3.8. Wavelet Neural Network (WNN) for Renewable Energy Forecasting

WNN is a kind of NN that combines the mathematical concept of wavelets with ANNs. Wavelets are mathematical functions that can be used to analyze and decompose complex signals or data into simpler components. They are particularly useful in analyzing signals with both time and frequency components, such as audio, images, and time-series data. Wavelets are employed as the activation functions for the hidden neurons in a WNN. The neurons are fed with input signals that have been broken down into wavelet coefficients. They then perform linear transformations on these coefficients to extract characteristics or features from the input information [160]. The outcome of each hidden neuron is then passed through a wavelet activation function, which applies a wavelet function to the transformed coefficients to produce an output signal. The use of wavelet coefficients and activation functions in a network enables it to effectively capture the time and frequency features of the input data. As a result, this approach is highly advantageous for tasks that involve time series forecasting, signal processing, and image analysis [161]. WNNs can capture non-linear correlations between input and output data as well as manage noisy or missing data. They are also computationally efficient and can be trained using standard back-propagation techniques [162,163].

The application of WNN in renewable energy forecasting is attributed to its capability to capture the temporal and frequency characteristics of the input data. For example, a study by Dewangan et al. (2015) proposed using wavelet neural networks (WNN) with Levenberg-Marquardt training for short-term solar irradiance forecasting. WNNs employ adaptive wavelet-based activation functions, resulting in better accuracy and generalization capability than conventional sigmoidal neural networks [164]. Chitsaz et al. (2015) developed a new engine for wind power prediction using a wavelet neural network with multi-dimensional Morlet wavelets as activation functions. The model is optimized with an improved clonal selection algorithm and trained with the maximum correntropy criterion. Results using real-world data in Alberta, Canada, show the effectiveness of the proposed approach [165]. Shen et al. (2017) also developed a WNN-based technique for wind power prediction. They optimized the forecasting model using EKMOABC, a new multi-objective artificial bee colony (MOABC) technique. They found that the proposed model and algorithm produced higher-quality prediction intervals for wind power forecasting [166]. Sharma et al. (2016) created a mixed WNN using the wavelet transform, which outperformed other methods for short-term solar irradiance forecasting in Singapore [167]. The use of WNN models in conjunction with other ML models has been implemented to enhance the precision of forecasting. For instance, Hamed H.H. Aly (2020) created a series of hybrid DL clustered models utilizing various AI systems such as RKF, FS, WNN, and ANN to predict wind speed and power with remarkable accuracy. They proposed and tested twelve distinct models, and it was found that the clustered model, which combined WNN and RKF, yielded the most optimal results [161].

### 3.9. RBNN for Renewable Energy Forecasting

The radial basis function neural network (RBNN) is an ANN type that has gained extensive usage in different fields, including but not limited to pattern recognition, control systems, time-series prediction, and function approximation [168]. In RBNN, the neurons in the hidden layer are typically implemented using radial basis functions (RBFs). RBFs are a class of functions that have a center, which represents a prototype or a reference point, and a width, which controls the influence of the function on the input space [169,170]. In RBNN, the input vector is first transformed by the hidden layer using the RBFs, and the resulting outputs are then combined linearly to produce the final output. The weights of the linear combination are typically learned using a supervised learning algorithm such as backpropagation [169,170]. RBNN can approximate any continuous function with a

high degree of accuracy, provided that an ample number of hidden neurons are available. Additionally, RBNN exhibits strong generalization performance when the RBF centers are thoughtfully selected to represent the input data distribution, particularly for unseen data [170].

RBNN has been applied in renewable energy forecasting due to its ability to capture the nonlinear relationships and temporal dependencies in the input data. In 2011, Wu et al. introduced a wind power prediction model based on RBNN that can forecast one hour ahead. They preprocessed the samples using the Grubbs test and evaluated the accuracy of their forecasting results by comparing them with the actual wind power outputs. The study demonstrated that the method proposed by the authors can provide reliable and consistent predictions [171]. Chala et al. (2015) created an improved T-RBNN to predict wind speed in areas with no measurements. The model was tested and validated with nearby wind stations, demonstrating acceptable accuracy [172]. M. Madhiarasan (2020) also developed a recursive RBNN to predict wind speed reliably utilizing wind speed, temperature, and wind direction inputs. According to the author, the model can improve the management, control, and protection of power systems, and the simulation results show more accuracy compared to existing forecasting models [173]. RBNN models have also been used in combination with other ML models, such as SVR, to improve forecasting accuracy. For example, Ramedani et al. (2014) utilized four artificial intelligence models to predict global solar radiation in Tehran, Iran. They utilized SVR with polynomial and radial basis neural network (RBNN) kernel functions. The combination of SVR and RBNN yielded the most accurate predictions compared to the other models [174].

### 3.10. GRNN for Renewable Energy Forecasting

The generalized regression neural network (GRNN) is frequently utilized for regression tasks that require forecasting continuous quantities [175]. A GRNN's fundamental structure comprises four layers, namely the input layer, pattern layer, summation layer, and output layer (Figure 9). In the input layer, the network takes in the input data, which could be a vector or a set of vectors. The pattern layer is where the network compares the input data to stored prototypes and computes the similarity between them. The prototypes are essentially a set of reference vectors that the network uses to make predictions. After receiving the output from the pattern layer, the summation layer calculates a weighted sum of the values using similarity values determined by the former. Finally, the output layer generates a continuous prediction for the network. GRNNs are trained using a process called radial basis function (RBF) learning [162]. Throughout the training process, the network modifies the prototypes and their corresponding weights to decrease the disparity between the anticipated and factual values of the training dataset. GRNNs have several advantages over other types of neural networks [176]. They are relatively easy to train and can be trained on small datasets. They are also efficient at making predictions and can handle noisy and incomplete data. Nevertheless, they may not demonstrate optimal performance when dealing with complex datasets that possess high-dimensional input spaces, and their predictions may be comparatively less accurate than those of other types of neural networks on such datasets.

GRNNs have been applied in renewable energy forecasting due to their ability to handle noisy and nonlinear data [177,178]. For example, Tu et al. (2022) propose a grey wolf optimization (GWO)-based GRNN for predicting energy output in solar power systems, which is affected by unpredictable external factors such as weather and cloud cover. They used wind speed, temperature, humidity, rainfall, and solar irradiance as inputs. A self-organizing map (SOM) is used for weather clustering and GRNN training with GWO. The proposed model achieves high prediction accuracy with shorter computational times and enhances the effective operation of solar power systems [179]. In 2021, M. Sridharan put forward a GRNN model that utilizes seasonal and meteorological factors such as months, latitude, longitude, altitude, clearness index, temperature ratio, and mean duration of sunshine per hour as input parameters to forecast global solar irradiance. The

outcomes demonstrate that this model offers precise predictions and is on par with other ML models such as fuzzy logic and ANN models. The average percentage error for the proposed GRNN model stands at 3.55%, whereas for fuzzy logic and ANN models, it is 4.64% and 5.49%, respectively [180]. Likewise, G. Kumara and H. Malikb (2016) proposed wind speed prediction in 67 Indian cities employing GRNN and multi-layer perceptron (MLP) models. Input variables included latitude, longitude, air temperature, daily solar radiation- horizontal, relative humidity, elevation, Earth temperature, cooling degree-days, atmospheric pressure, and heating degree-days. Compared to MLP, GRNN exhibited superior performance, achieving accuracy rates of 99.99% and 97.974% during training and 98.85% and 95.23% during testing, respectively [181].

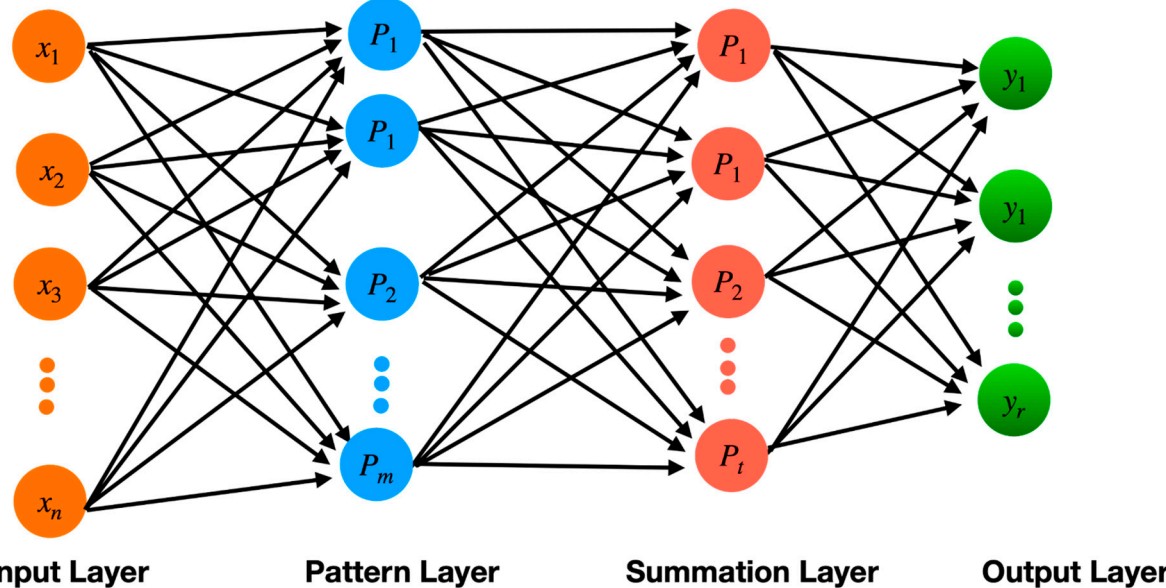

**Figure 9.** The schematic representation of GRNN [177].

### 3.11. ELM for Renewable Energy Forecasting

Extreme learning machines (ELM) were presented as a substitute for conventional gradient-based NN. ELM is designed to be computationally efficient and easy to implement while still providing high accuracy in various applications, including regression, classification, and clustering. ELM is composed of a solitary layer of neurons that are hidden. In this layer, the connections between the input layer and the hidden layer are generated randomly and remain constant. The output layer in ELM is typically a layer that performs linear regression or classification, and the connections between the hidden layer and the output layer are determined through analytical means using matrix inversion [182]. During the training stage, the input information is fed through the randomly generated hidden layer weights to produce a set of outputs, which are then used to calculate the output layer weights using matrix inversion. Once the output layer weights are determined, the model becomes capable of making predictions for new input data. ELM offers a significant benefit in terms of its fast training time. This is due to the fact that the weights of the hidden layer are randomly generated and remain fixed throughout the training process, which eliminates the need for any adjustments. This means that ELM can handle large datasets and complex problems with high-dimensional input spaces in a relatively short amount of time. Additionally, ELM is less prone to overfitting compared to traditional gradient-based neural networks, which can be beneficial in many applications. Overall, ELM is a simple and efficient NN model that can achieve high accuracy in a variety of applications, especially when dealing with large datasets and high-dimensional input spaces [183].

ELM models are utilized for forecasting renewable energy sources (RES), including wind and solar power [184]. For example, Shamshirband et al. (2015) utilized the extreme learning machine (ELM) for predicting horizontal global solar radiation (HGSR) using three different input parameter types. The ELM was compared to SVM, genetic programming (GP), and ANN, and found to provide higher accuracy and reliability, especially with multiple parameter-based estimations [184]. Golestaneh et al. (2016) also proposed a nonparametric approach using an extreme learning machine (ELM) for solar power generation forecasting. The approach generates fast, short-term predictive densities, achieving skillful and reliable probabilistic forecasts with fast frequency updates. Results show that ELM outperforms four probabilistic benchmarks in terms of accuracy and computational efficiency for two different solar power generation sites [185]. Likewise, Li et al. (2016) suggested an ELM and error correction model to precisely predict short-term wind power. The addition of an error correction model enhanced the accuracy of ultra-short-term wind power forecasts [186]. Hou et al. (2018) suggest a forgetting factor (FOS)-ELM model with a variable forgetting factor to predict solar radiation. Using the Bayesian Information criterion (BIC), they build and evaluate seven input combinations, with the FOS-ELM model showing improved RMSE and MAE compared to the classical ELM model. The study confirms FOS-ELM's effectiveness in daily global solar radiation simulation [187]. Likewise, Li et al. (2019) developed an ELM model to predict wind power with kernel mean p-power error loss to overcome the limitations of traditional BP neural networks. The method eliminates redundant data components using PCA and achieves lower prediction errors without compromising accuracy [188].

### 3.12. Ensemble Learning for Renewable Energy Forecasting

A ML approach called ensemble learning (EL) combines several models to provide more accurate predictions [189]. The idea is to train several models independently on the same data and then combine their predictions to make a final prediction. EL is particularly useful when a single model is not able to achieve high accuracy or when there is significant noise or variability in the data [190]. Bagging, boosting, and stacking are among the various types of EL techniques available [191,192]. Bagging involves training multiple models on various subsets of the training data with replacement. The final prediction is generated by combining the predictions of all the models. This technique is particularly useful when the base model is prone to overfitting [193]. Boosting involves training multiple models in a sequence where each subsequent model aims to correct the errors of its predecessor. The ensemble prediction is generated by aggregating the predictions of all the models. Boosting is particularly useful when the base model is prone to underfitting. In contrast, stacking involves training multiple models and using their predictions as input to a higher-level model that learns how to combine them [191]. Stacking is particularly useful when the base models have different strengths and weaknesses. EL has proven to be useful in a variety of applications, including image classification, natural language processing, and recommendation systems. However, it can be computationally expensive and requires careful tuning of the ensemble parameters to achieve optimal performance [194].

The bagging technique is a popular EL method utilized in the prediction of renewable energy [193]. For example, Guia et al. (2020) conducted a study where a bagging-based EL technique was applied to forecast solar irradiance using weather patterns. The base learner in the ensemble was a pre-processed stacked LSTM model. The study showed that bagging-based ensemble learners outperformed individual learners in terms of accuracy, as evidenced by multiple metrics [195]. Another EL method used in renewable energy forecasting is the boosting technique. P. Kumari and D. Toshniwal (2021) suggested an ensemble model for estimating hourly global horizontal irradiance that integrates extreme gradient-boosting forests and deep neural networks (XGBF-DNN). They used a framework that incorporates feature selection and variety in base models, and the suggested model outperforms other models in prediction error with a forecast accuracy score range of 33–40%, making it a reliable and appropriate model for solar energy system planning

and design [196]. Al-Hajji et al. (2021) reported a comparative analysis of stacking-based ensembles for predicting solar radiation one day ahead using ML. They explored three stacking methods (feed-forward NN, SVR, and k-nearest neighbor) for merging base predictors, and assessed their performance over a year. Their findings revealed that stacking models, which combine heterogeneous models utilizing neural meta-models, outperformed recurrent models [197]. Wind energy is also forecasted using ensemble learning. For instance, Banik et al. (2020) created an accurate wind speed and wind power prediction methodology using ensemble machine learning algorithms. The probabilistic nature of wind power production makes it challenging to balance supply and demand in power systems. The proposed approach helps minimize the need for auxiliary energy balancing and reserve power [198]. Gomes da Silva et al. (2021) also developed a novel decomposition-ensemble learning approach to forecast wind energy using complete ensemble empirical mode decomposition (CEEMD) and stacking-ensemble learning based on machine learning algorithms. The proposed model outperforms single and conventional models, with a performance improvement ranging from 0.06% to 97.53%, making it an efficient and accurate model for wind energy forecasting [199].

### 3.13. Transfer Learning (TL) for Renewable Energy Forecasting

TL is a ML approach in which a previously trained model is utilized as a reference point for a new task rather than developing a new model from scratch. Typically, the pre-trained model has been trained on a large dataset and has learned useful features that can be applied to other related tasks. The TL process involves taking the pre-trained model and fine-tuning it on a new dataset for the new task. The fine-tuning process entails modifying the pre-trained model by adding new layers or modifying existing layers to fit the new data. This approach assists in enhancing the efficacy of the pre-trained model for the new task without the need to train a new model from scratch (Figure 10) [200]. TL has numerous benefits, one of which is that it uses less data and computer resources while training new models. It also enables the use of pre-trained models that have already learned features that are relevant to the new task, which can lead to better performance than training a new model from scratch [201]. Several areas of ML have made extensive use of TL, including computer vision, natural language processing, and speech recognition. TL has been employed in computer vision to increase the performance of picture recognition tasks, including object detection, classification, and segmentation. TL has also been employed in natural language processing to increase the performance of tasks such as sentiment analysis, language translation, and named entity recognition [202].

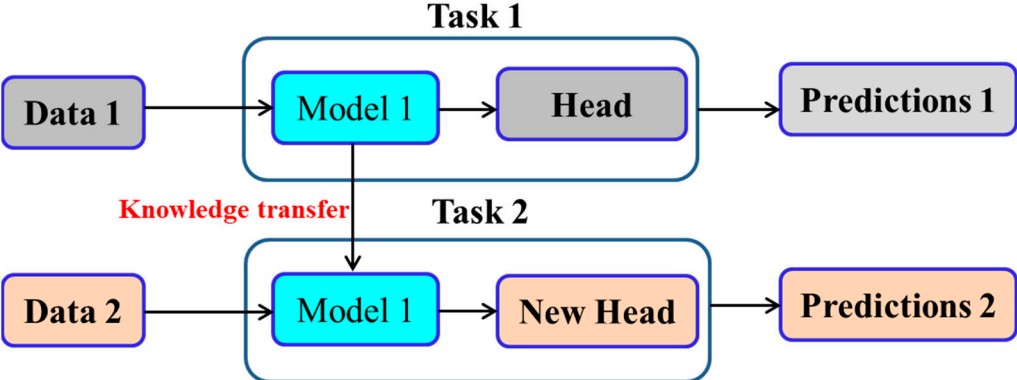

**Figure 10.** The schematic representation of TL.

In renewable energy forecasting, TL has been utilized to enhance the accuracy of renewable energy forecasts by leveraging knowledge learned from related tasks. One common approach to TL in renewable energy forecasting is to use pre-trained models from other related forecasting tasks. For example, Sarmas et al. (2022) propose using TL with stacked LSTM models to accurately forecast solar plant production in situations where

there is a lack of data. They compared three TL models against a non-TL model and a smart persistence model, with TL models achieving significant improvements in accuracy. They conclude that TL is an effective tool for power output forecasting, particularly for newly constructed solar plants, intending to achieve energy balance and manage demand response [203]. Hu et al. (2016) also developed deep neural networks trained on data-rich wind farms to extract wind speed patterns and transfer this information to newly built farms, significantly reducing prediction errors [204].

*3.14. Hybrid Model (HM) for Forecasting Renewable Energy*

HMs in renewable energy forecasting are ML models that combine multiple techniques, including ANN, SVM, and statistical models, to improve the accuracy of predictions [202]. HMs offer several advantages over individual models by leveraging the strengths of each technique while compensating for their limitations [205]. In renewable energy forecasting, HMs can be employed to enhance prediction accuracy and overcome some of the limitations of classic ML models. For example, traditional models may struggle with capturing the complex and nonlinear relationships between RES and their influencing factors. HMs can address this challenge by combining multiple models and techniques to capture a wider range of features and enhance forecast accuracy [205].

One example of a HM used in renewable energy forecasting is the CNN and LSTM model. In a recent paper, Lim et al. (2022) proposed the HM of a LSTM and CNN to accurately forecast the power generation of photovoltaic (PV) systems [125]. The model first classified weather conditions using CNN and then learned power generation patterns using LSTM. The suggested model's mean absolute percentage error was 4.58 on sunny days and 7.06 on cloudy days, indicating the possibility of precise power generation forecasting and optimization of PV power plant operations [125]. Another study by Mbah et al. (2013) employed a HM for short-term power prediction for a photovoltaic plant. The model combines SARIMA and SVM methods and is tested on a 20 kWp GCPV plant. Results show good accuracy and outperform both SVM and SARIMA models [206]. Eseye et al. (2018) employed HMs to forecast power for a real microgrid PV system over the short term (one day in advance). In terms of predicting accuracy, the model, which integrates wavelet transform (WT), particle swarm optimization (PSO), and SVM approaches, surpasses seven other methods [207]. Likewise, H. Aly (2020) used HMs to improve wind speed forecasting for better renewable energy integration. The proposed models combine WNN and ANN, time series (TS), and recurrent Kalman filter (RKF) techniques. The best-performing models, validated using unseen datasets, are WNN, TS, and RKF, in that order [208]. Table 2 summarizes various studies that investigated applying ML and DL models to forecast or predict renewable energy sources such as wind and solar power.

**Table 2.** Literature related to solar and wind power forecasting using ML and DL techniques.

| Algorithms Used | Application | Inputs Used | Prediction Outputs | Ref. |
|---|---|---|---|---|
| ANN and regression models (LR, M5P, DT, and Gaussian process regression (GPR)) | Solar Energy | Solar irradiance, ambient temperature, relative humidity, PV surface temperature, wind speed, and dust on PV panels. | The hourly power output of the PV system | [209] |
| MARS, CART, M5, and random forest | Solar Energy | Minimum, maximum, and average temperature, wind speed, rainfall, dew point, GSR, atmospheric pressure, and solar azimuth | One to six days' worth of hourly solar radiation | [22] |
| ANN | Solar Energy | Pressure, relative humidity, wind speed, ambient temperature, and sunshine duration | Monthly average daily GSR | [20] |
| ANN, DBN, autoencoder, and LSTM | Solar Energy | Sunshine hours, daily average solar irradiation, location, temperature, etc. | Solar power | [210] |
| SVM, ANN, DL, kNN | Solar Energy | Local time, temperature, pressure, wind speed, relative humidity, and past time-series solar radiation | Hourly solar radiation | [23] |

**Table 2.** *Cont.*

| Algorithms Used | Application | Inputs Used | Prediction Outputs | Ref. |
|---|---|---|---|---|
| ANN | Solar Energy | Global solar irradiance, direct beam solar irradiance, time, and power generated from the solar PV | Forecasting solar power | [211] |
| Deep CNN | Solar Energy | Longitude, latitude, time, and altitude; humidity, temperature, wind velocity, moisture, etc. | Solar power predictions | [212] |
| ANN, kernel ELM | Wind Energy | Wind speed | Short-term wind speed forecasting | [30] |
| BP network, RBF network, and NARX models | Wind Energy | Time series historical weather data for three years in 15 min intervals: Wind direction and speed, radiation, temperature, reflected radiation, humidity, etc. | Prediction of wind speed | [213] |
| 2D-CNN | Wind Energy | Historical wind speeds | Twenty-four-hour forecasting of wind speed | [15] |
| LSSVM, HM, LMD | Wind Energy | Five short-term wind speed datasets | Short-term wind speed prediction | [33] |
| LASSO, kNN, RF, XGBoost, SVR | Wind Energy | Daily wind speed, daily standard deviation, and daily wind power | Long-term wind power forecasting | [29] |

## 4. Challenges and Future Prospects

For predicting the output of RES such as solar and wind, there are several ML and DL algorithms. Each method has its benefits as well as drawbacks. The best ML or DL method for predicting RES depends on the specific application and available data. Linear regression can be a good choice as a baseline model, while random forest, xgboost, and SVMs are suitable for handling non-linear relationships and complex data. The variant recurrent neural networks (RNNs) models in DL are particularly useful for time-series forecasting but can be computationally expensive to train relative to classical ML models. The variability of wind and solar poses a significant challenge to ensuring a reliable and steady electricity supply that meets demand. ML and DL methods for renewable energy forecasting face the challenge of intermittency, in which energy output varies based on environmental factors such as weather and time of day. Traditional forecasting methods relying on historical data and assumptions may not be sufficient for capturing the complex and dynamic nature of renewable energy sources. To address this, advanced ML and DL techniques that learn from real-time data and adapt to changing conditions can provide a potential solution. For instance, RNNs and LSTMs neural network models have demonstrated efficacy in forecasting renewable energy output using real-time weather data. These models capture complex relationships between environmental factors and energy output and make accurate predictions, even in the face of variability and unpredictability. Classical ML models can also be used for forecasting, but they may not perform as well as specialized time-series forecasting methods such as autoregressive models, moving averages, and RNNs [214,215]. The main reason for this is that classical ML models frequently assume independent and uniformly distributed data points, which is often not the case with time series data. Time series data is characterized by temporal dependencies, meaning that the values at one time point are influenced by the values at previous time points. The assumption that the input variables are independent of each other makes it challenging for classical ML models to grasp the patterns and trends in the data. Furthermore, classical ML models are not optimized for handling time-varying features or unevenly spaced time-series data, which are common in time-series forecasting. For example, a classical ML model may not be able to capture seasonality or trends in the data that occur over long periods.

For handling unevenly spaced time series data and identifying patterns and trends to produce precise forecasts, specialized time-series forecasting techniques such as autoregressive models, moving average models, and RNNs are the best options [216]. Hybrid models are becoming increasingly popular for forecasting RESs, as they can combine tradi-

tional time-series analysis with ML algorithms to improve accuracy and reduce the risk of overfitting or underfitting. RNNs, in particular, are specifically designed to handle time-series data and can capture temporal dependencies to learn from long-term patterns and trends, making them a popular choice for forecasting renewable energy sources. These models have the potential to enhance the operation and development of renewable energy systems as well as their grid integration. However, forecasting RES remains challenging due to their variability and unpredictability, highlighting the need for ongoing research and development of advanced forecasting techniques.

ML and DL models offer promising prospects for forecasting renewable energy sources. These models can provide more accurate predictions by processing large amounts of data and detecting complex patterns that humans may miss. These models also enable real-time forecasting to adapt to changing weather conditions, improving grid stability and enabling better decision-making. Accurate predictions also support better resource planning, leading to more efficient operations and cost savings. Furthermore, improved predictions can facilitate the integration of RES into the grid, reducing instability and enhancing overall grid performance. With careful consideration of data quality, model complexity, and validation, the use of ML and DL models has significant prospects for optimizing renewable energy operations and improving grid stability.

## 5. Conclusions

Renewable energy sources, such as wind and solar power, are becoming increasingly important for meeting the world's energy needs. However, their variability and unpredictability pose significant challenges for energy system operators. Accurate forecasting of renewable energy generation is critical for ensuring the stability and reliability of the grid. ML and DL algorithms have emerged as promising tools for renewable energy forecasting. This review provides an overview of the current state of the art and prospects of ML and DL algorithms for renewable energy forecasting. Classical ML models, such as linear regression, have been widely used for renewable energy forecasting and can be a good choice for a baseline model. It is simple, easy to interpret, and requires fewer computational resources. However, it may not be able to capture the non-linear relationships and complex patterns in the data. Random forest, SVMs, and XGBoost models have been shown to perform better than linear regression for renewable energy forecasting. These models can handle non-linear relationships and complex data and can provide accurate predictions even in the presence of noise and outliers. However, specialized time-series forecasting methods such as autoregressive models, moving averages, and RNNs are ideal for handling unevenly spaced time series data and capturing patterns and trends to provide accurate predictions.

Hybrid models that combine traditional time-series analysis with ML and DL algorithms have also been used for renewable energy forecasting. These models can capture both the linear and non-linear relationships in the data and can provide accurate predictions even in the presence of noise and outliers. However, the design of hybrid models is a challenging task, as it requires a good understanding of both time-series analysis and ML/DL algorithms. Despite the progress made in renewable energy forecasting with ML and DL algorithms, there are still some challenges that need to be addressed. One of the main challenges is the lack of high-quality data for training and validation. The data for renewable energy forecasting is often sparse, noisy, and incomplete, which makes it difficult to build accurate models. Another challenge is the lack of transparency and interpretability of ML and DL models. Many ML and DL models are black boxes, which makes it difficult to understand how they make predictions. There are several opportunities for future research in the field of renewable energy forecasting with ML and DL algorithms. Further research is required in the development of models that can handle multiple renewable energy sources simultaneously. Another area is the development of models that can handle the uncertainty and variability of renewable energy sources. The integration of weather data, grid data, and other external data sources can also improve the accuracy of renewable energy forecasting.

**Author Contributions:** Conceptualization, N.E.B., M.D.C. and A.G.S.; writing original draft preparation, N.E.B., M.D.C. and A.G.S.; writing review and editing, N.E.B., M.D.C. and A.G.S. All authors have read and agreed to the published version of the manuscript.

**Funding:** The authors declare that no funds, grants, or other support were received during the preparation of this manuscript.

**Institutional Review Board Statement:** Not applicable.

**Informed Consent Statement:** Not applicable.

**Data Availability Statement:** The analyzed datasets during this study are available from the corresponding author upon reasonable request.

**Conflicts of Interest:** The authors declare no conflict of interest.

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
