# Peer review of "Forecasting Renewable Energy Generation with Machine Learning and Deep Learning: Current Advances and Future Prospects"

_sustainability, doi:10.3390/su15097087_

Round 1

Reviewer 1 Report

The paper titled "Forecasting Renewable Energy Generation with Machine learning and Deep Learning: Current Advances and Future Prospects" is reviewed.

1. The authors organized the paper well and discussed about the ML and DL techniques for predicting the RES and its integration in to the grid. However, the paper requires a certain revisions.

2. The literature discussed is not enough and to drag the attention of the readers, the most appropriate literature must be indicated in the tabular form as critical literature review.

3. The author must discuss about the various ML and DL algorithms along with the ANN training techniques so far available in the literature.

4. The paper must be checked for journal template and the font sizes are irregular in some sections.

5. Need significant revisions in the conclusion section, at present it is not sound. Also, discuss about the effective ML and DL techniques that so far not available.  The possible challenges and opportunities in this research area with the DL and ML methods for future purpose must be illustrated in the point by point manner.   

Reviewer 2 Report

The cross-sectional article presents the strengths and weaknesses of various methods based on artificial neural networks in forecasting renewable energy generation. Both artificial neural networks and renewable energy sources are becoming increasingly popular. Hence their combination seems to be a natural consequence. The discussed topic may be of great importance in the future for the implementation of energy flow management processes in the power system in the context of continuity of its supply and quality. In addition, it can be extended to forecast emergencies, which will further increase energy security. The way of presenting the issues mentioned above in the article is clear, well illustrated with graphic material, and properly documented with references to the bibliography.

Reviewer 3 Report

The major comments and questions for the paper are as follows. 

1.      Please merge the first and fourth sentences and move them after the motivation.

2.      Introduction, please discuss the physical models used in renewables forecasting and their limitations.

3.      The literature review focuses on solar radiation and wind energy forecasting. Please add more work about solar energy forecasting.

4. In Section II, please add a summary paragraph at the beginning.

5.      It is suggested to reorganize Section II as follows.

2.1. Supervised Learning

2.2. Unsupervised Learning

2.3 Reinforcement Learning

2.4. Deep Learning

6.      When introducing regression, GRNN, and ANN, please indicate input variables used for forecasting.

7.      When introducing Reinforcement Learning (RL) for forecasting renewables, the reviewed work is to develop an operation control strategy, not power forecasting. Since RL cannot be directly used for forecasting, it is suggested to review RL to select input variables for forecasting.

8.      It is suggested to review existing work based on extreme learning machines (ELM), Wavelet neural networks (WNN), and radial basis neural networks (RBNN).

9.      The review of Ensemble Learning is only on solar radiation forecasting. It is suggested to review more about solar and wind power forecasting.

10.   In the challenges and future prospects section, please discuss the main challenge of renewables (intermittency).

Some minor comments are as follows. 

1.      Introduction, the font size of the first paragraph is bigger than that of the other paragraphs.

2.      The last layer in Figure 9 should be the output layer.

3.      When introducing the hybrid models, one piece of work was already mentioned in section 2.1.1

4.      There are a lot of different font sizes in the same section (some sentences in the introduction, the last sentence in section 2.1, etc.)

5.      There are lots of spaces between words. Please review the entire paper carefully.

Reviewer 4 Report

The authors have worked on "Forecasting Renewable Energy Generation with Machine learning and Deep Learning: Current Advances and Future Prospects". The work seems interesting and can be considered after a few modifications.

Introduction: The section is well written however the novelty of the manuscript needs to be added along with recent literature.

Author need to add comparison tables of the " all ML and DL Technics with pros and cons in different applications point of view"

All topic titles are 2.1.1,need to change 

Round 2

Reviewer 3 Report

Most of the comments have been well addressed. The format of new figures 6-9 is not consistent. Please try to use the same format.

Author Response

We appreciate the reviewer's insightful comments and have addressed them in the revised manuscript